# Biological Response Evaluation of Human Fetal Osteoblast Cells and Bacterial Cells on Fractal Silver Dendrites for Bone Tissue Engineering

**DOI:** 10.3390/nano13061107

**Published:** 2023-03-20

**Authors:** Domenico Franco, Antonio Alessio Leonardi, Maria Giovanna Rizzo, Nicoletta Palermo, Alessia Irrera, Giovanna Calabrese, Sabrina Conoci

**Affiliations:** 1Department of Chemical, Biological, Pharmaceutical and Environmental Sciences, University of Messina, Viale Ferdinando Stagno d’Alcontres 31, 98168 Messina, Italy; 2Department of Physic and Astronomy, University of Catania (Italy), Via Santa Sofia 64, 95123 Catania, Italy; 3CNR IMM, Catania Università, Via Santa Sofia 64, 95123 Catania, Italy; 4CNR URT Lab SENS, Beyond NANO, Viale Ferdinando Stagno d’Alcontres 31, 98166 Messina, Italy; 5Department of Chemistry “Giacomo Ciamician”, University of Bologna, 40126 Bologna, Italy

**Keywords:** Ag dendrites, cytocompatibility, antibacterial property, orthopedic devices, bone tissue engineering

## Abstract

Prosthetic joint replacement is the most widely used surgical approach to repair large bone defects, although it is often associated with prosthetic joint infection (PJI), caused by biofilm formation. To solve the PJI problem, various approaches have been proposed, including the coating of implantable devices with nanomaterials that exhibit antibacterial activity. Among these, silver nanoparticles (AgNPs) are the most used for biomedical applications, even though their use has been limited by their cytotoxicity. Therefore, several studies have been performed to evaluate the most appropriate AgNPs concentration, size, and shape to avoid cytotoxic effects. Great attention has been focused on Ag nanodendrites, due to their interesting chemical, optical, and biological properties. In this study, we evaluated the biological response of human fetal osteoblastic cells (hFOB) and *P. aeruginosa* and *S. aureus* bacteria on fractal silver dendrite substrates produced by silicon-based technology (Si_Ag). In vitro results indicated that hFOB cells cultured for 72 h on the Si_Ag surface display a good cytocompatibility. Investigations using both Gram-positive (*S. aureus*) and Gram-negative (*P. aeruginosa*) bacterial strains incubated on Si_Ag for 24 h show a significant decrease in pathogen viability, more evident for *P. aeruginosa* than for *S. aureus*. These findings taken together suggest that fractal silver dendrite could represent an eligible nanomaterial for the coating of implantable medical devices.

## 1. Introduction

The repair of bone defects is a well- known physiological process implicating numerous cell varieties and signaling molecules cooperating on the damaged site to restore bone tissue [1]. Nevertheless, when large bone defects occur due to trauma, degenerative diseases (osteoarthritis), inflammatory diseases (rheumatoid arthritis), and tumor resection, natural healing is compromised, and it is necessary to remedy this with existing bone graft strategies [2]. Although autologous bone grafting currently defines the gold standard method for these injuries, many disadvantages limit its use, including the associated pain, donor-site morbidity, and the risk of infection. The allograft represents a typical alternative to an autologous bone graft, even considering the other problems that can be associate with it, including immunogenic reactions and the possibility to transfer diseases from donors to patient [3]. Bone restoration is a complex physiological process that combines mechanical strength and revascularization with osteogenesis, osteoconduction, and osteoinduction [4]. Therefore, the ideal bone substitute should satisfy specific requirements, such as biocompatibility, the avoidance of the probability of immunogenic rejection and/or disease infection, and the ability to integrate into the host bone and regenerate it [5]. In addition, it should be osteoconductive, osteoinductive, and resorbable [6].

In orthopedic surgery, prosthetic joint replacement is the most used approach to repair function and reduce pain in damaged joints, even if it is often associated with prosthetic joint infection (PJI). PJI is a harmful complication associated with patient morbidity and considerable economic impact on the healthcare system. PJI is mainly due to the metal surface of the implant that offers an ideal environment for the bacteria growth and biofilm formation [7]. The bacteria most commonly involved in PJI are *Staphylococcus aureus* (*S. aureus*), *Pseudomonas aeruginosa* (*P. aeruginosa*), or *Escherichia coli* (*E. coli*), mainly due to their high virulency and increased drug resistance. All these events together trigger a strong inflammatory reaction leading to bone-damaging processes which cause the mobilization of the prosthesis, and the progressive destruction of the whole joint [8]. In Italy, prosthetic infections occur in about 2–5% of the cases undergoing primary arthroplasty, and the cost for an infected prosthesis revision is 4.8 times that of a primary implant [9]. Antibiotic treatments of bone infections have been reported to fail unless they are associated with the whole suppression of microorganisms from the surgical site. Therefore, prevention, and finally, complete bacteria eradication from the implantation site is necessary for the success of the graft [10]. In this context, in recent years, several approaches have been proposed to resolve the problem of PJI and to reduce the biofilm formation, such as the coating of implantable devices with a nanostructured antibacterial coating [11,12]. Several advanced nanomaterials displaying antibacterial activity, including metal ions and metal oxide nanoparticles, have been broadly reported in the literature [13,14]. Among these, extensive studies have been reported for silver [15], copper [16], gold [17], titanium dioxide [18], zinc [19], and zinc oxide [20]. The antibacterial properties of these metal ions and metal oxide NPs are mainly due to their ability to release metal ions in aqueous media, resulting in extremely toxicity against bacteria [21]. In particular, silver nanoparticles (AgNPs) have gained great interest due to their ability to reduce bacterial adhesion. Several in vitro studies have been performed to verify the antimicrobial activity of AgNPs, but although these have shown remarkable anti-biofilm properties, they have also showed cytotoxic effects, limiting their use in the biomedical field [22]. Currently, several researchers are trying to define the most appropriate concentration, size, and shape of silver nanoparticles that should be used for the coating of implantable devices [23]. In particular, in recent years some researchers have focused their attention on the antibacterial properties of different Ag nanostructures, including nanoflowers and nanodendrites, due to their interesting chemical, optical, and biological properties [24]. Silver nanoflowers have been employed for use as biosensors [25], nanodevices [26], and SERS substrates [27], displaying optimized properties compared to their nanoparticles. On the other hand, dendritic nanostructures have gained great attention due to their antibacterial properties as antibacterial coatings for medical implants [28,29,30,31].

In the literature, Ag dendrite fabrication is well documented with several approaches using electrochemical or chemical synthesis by using template or even templateless methods [32]. Ag dendrites can be obtained by the use of template methods involving the reduction of Ag salts on a pre-patterned substrate, i.e., porous spongy nickel substrates or polymeric templates [33]. Template methods favor a more controlled fabrication in terms of size and shape of the Ag dendrites at the cost of longer synthesis time (up to days) and the common use of a supplementary energy supply in the process using photoactivation [34], ultrasound [35], microwaves [36], voltage [37], or catalysts [38]. Moreover, the template elimination implies the use of other chemical processes by using acid solutions of HF or HCl that may have detrimental effects on the dendrite morphology or introduce contaminations from the template. Templateless methods, at the price of a less control regarding shape and size, permits a fabrication protocol involving fewer chemical steps and a simpler route. Considering the electrochemical route, Ag dendrites can be fabricated by monitoring the Ag reduction on the substrate electrode. This method permits the regulation of dimension and shape by changing the solution, pH, reaction time, and the applied bias [37], and the elongation of the Ag trenches can be well controlled by varying the growth time between tens of minutes to a few hours [35]. Apart from the common electrochemical downsides of costly Pt counter electrodes and a conductive substrate for their growth, other drawbacks are related to the common synthesis of asymmetrical dendrites, the lack of Ag robustness, and the general morphological instabilities after the bias removal due to potentiostatic transitions [39]. Other templateless methodologies involving the casual deposition of Ag ions from salts reduction can be found in the literature. Commonly, due to the weak reducing agent, the reduction times are long, involving synthesis processes from hours to days, and several additives, such as citrates, p-aminoazobenzene [40], sodium borohydride [41], or Zn microparticles [42], can be used to reduce the reaction time. To overcome the poor dimension and shape control of this method, specific capping agents (i.e., Polyvinylpyrrolidone) can be used during growth [43]. However, for several applications, in addition to the antibacterial functions, the use of capping agents is undesirable, since it limits the effectiveness of the Ag activity [44].

In this study, we assessed the biological response of human and bacterial cells on fractal silver dendrite-coated substrates for use in bone tissue engineering applications Italian patent application N. 102022000020145 [45]. Ag dendrites have been produced by industrially compatible silicon-based technology [46,47] and have been synthesized without the use of capping agents nor templates, using previously demonstrated fractal morphology [48]. In vitro studies using hFOB (human fetal osteoblastic cells) and *P. aeruginosa* and *S. aureus* bacteria have been performed to assess the cytotoxic effect and antimicrobial activity of these substrates.

## 2. Materials and Methods

### 2.1. Chemicals and Materials

Silver nitrate (AgNO_3_) 0.05 N was acquired from Scharlau (Barcelona, Spain). Hydrofluoric acid (HF) 40% was purchased from Honeywell (PERLABO S.A.S., Catania, Italy). Acetone and Isopropanol were acquired from Sigma-Aldrich (Merk Life Science S.r.l., Milan, Italy). Deionized water (18 MΩ·cm) was obtained from Q-Millipore (Darmstadt, Germany) and it was used for all the aqueous solutions used in sample fabrication. Commercial 4′′ (100) oriented silicon wafers, with a 500 μm of thickness and p-type doping (1–5 Ω·cm), were purchased from Siegert Wafer (Aachen, Germany) and are used as substrates.

### 2.2. Ag Dendrites Structural Characterizations

Ag dendrites were fabricated by a metal assisted chemical etching (MACE) protocol based on a wet etching process driven by a metal catalyst deposited onto the surface of the silicon, involving a chemical bath of Si substrates in an AgNO_3_/HF (0.02 M/5 M) aqueous solution. This protocol is a widely diffused approach for the cost-effective realization of Si NWs [48]. The morphology of the samples, in cross-section and in plan-view, was investigated using a ZEISS Supra 25 scanning electron microscope (Zeiss, Oberkochen, Germany).

### 2.3. Cell Culture

The human fetal osteoblastic cell line (hFOB 1.19) used in this work was acquired from the American Type Culture Collection (ATCC, Manassas, VA, USA). Osteoblast cells were grown in Ham’s F12 Medium–Dulbecco’s Modified Eagle’s Medium (1:1) (Merk Life Science S.r.l., Milan, Italy), complemented with 10% Fetal Bovine Serum (FBS, Merk Life Science S.r.l., Milan, Italy), 2.5 mM L-glutamine (Merk Life Science S.r.l., Milan, Italy), 1% penicillin/streptomycin/amphotericin (Merk Life Science S.r.l., Milan, Italy), and 0.3 mg/mL G418 (Thermo Fisher Scientific,, Waltham, MA USA) and maintained in a humidified atmosphere containing 5% CO_2_ at 37 °C. The medium was substituted twice a week, and the cells were divided when 80% of confluence was reached.

### 2.4. Cell Viability and Proliferation on Fractal Silver Dendrites

A cell viability assay of the cells cultured on fractal silver dendrites was performed by MTT [3-(4,5-dimethylthia- zol-2-yl)-2,5-diphenyltetrazolium bromide] assay (Merk Life Science S.r.l., Milan, Italy). Specifically, 5 × 10^5^ hFOB, resuspended in 50 μL of growth medium, were seeded on the surface of the Si substrate coated with fractal silver dendrites (Si_Ag) and the untreated Si substrate (Si), in a 24-well plate and maintained in a humidified atmosphere containing 5% CO_2_ at 37 °C for 4 h. Subsequently, the growth medium was added to completely cover the fractal silver dendrites, and the plates were re-incubated, as above, for 24, 48, and 72 h. At the end of the incubation times, the medium from each well was eliminated and substituted with 200 μL of MTT solution (1 mg/mL in FBS-free medium) for 2 h at 37 °C and 5% CO_2_. Next, the MTT solution was eliminated, each well was rinsed twice with PBS (Merk Life Science S.r.l., Milan, Italy), and the produced crystals were dissolved in 200 μL of DMSO. Then, the optical density at 540 nm was read using a synergy HT plate reader (BioTek Instruments, Inc., Winooski, VT, USA).

The hFOB cell proliferation on Si_Ag was analyzed by DAPI staining, as previously reported in [49]. Briefly, for DAPI staining, hFOB cells grown on Si_Ag were fixed in 4% PFA (Paraformaldehyde) for 15 min (Thermo Fisher Scientific, Waltham, MA USA), rinsed three times in PBS, permeabilized in 0.3% Triton X-100 for 5 min, and the nuclei were counterstained with DAPI (1:5000) in PBS for 5 min. The sections were mounted with fluorescent mounting medium (PermaFluor Aqueous Mounting Medium, Thermo Fisher Scientific, Waltham, MA USA), and digital pictures were acquired using a Leica DMI4000B fluorescence microscope (three digital pictures/scaffold). Finally, the nuclei were calculated by Fiji Image J recognition software (ImageJ2 version:2.9.0/1.53t; Open source image processing software, http://imagej.net/Contributors, accessed on 19 February 2023). The differences in proliferation rates were calculated using a *t*-test. Each biological test was performed in triplicate for each experiment.

### 2.5. Bacterial Strain and Growth Conditions

*P. aeruginosa* (ATCC 27853 strain, LGC Promochem, Milan, Italy) and *S. aureus* (ATCC 29213 strain, LGC Promochem, Milan, Italy) were grown in Bertani Broth (LB, Sigma-Aldrich, Milan, Italy) and Tryptone Soya Broth (TSB, Sigma-Aldrich, Milan, Italy), respectively, at 37 °C under constant shaking (250 rpm) in an orbital shaker (ARGO lab SKO-D XL, Carpi, Italy). Both bacterial strains were conserved at −80 °C in their specific medium, complemented with 20% glycerol.

### 2.6. Antibacterial Assay

To evaluate the antibacterial activity, bacterial suspensions of both strains, in the exponential growth phase, were prepared by inoculating 3–4 colonies in fresh medium (LB for *P. aeruginosa* and TSB for *S. aureus*). Each inoculum was incubated at 37 °C under continuous shaking (250 rpm) in an orbital shaker (ARGO lab SKO-D XL, Carpi, Italy) until the 0.5 McFarland turbidity standard, corresponding to approximately 2 × 10^8^ CFU/mL, was reached. Then, bacterial suspensions were diluted to obtain 2 × 10^7^ cells/mL and 20 µL/cm^2^ was placed onto the surfaces of Si_Ag. Before use, the Si_Ag was pre-decontaminated by UV-C treatment (1 h per side). Si_Ag with bacteria was then placed in 24-well plates and maintained at 37 °C for 30 min in a humidified atmosphere. A silicon substrate, without fractal silver dendrites (Si), was used as the control. After the incubation period, 2 mL of fresh medium was supplemented at each well, and the plate was stored at 37 °C overnight. The next day, a colony forming units (CFU) assay was performed to quantify the number of viable cells. Particularly, 100 µL of microbial suspension was acquired from each replicate and serially diluted 1:10 (1 to 10 dilution) in PBS. Then, 100 µL from each dilution was distributed on a solid medium and stored overnight at 37 °C. The following day, colonies ranging from 30–300 were calculated to establish the number of CFUs in 1 mL:(1)$$\mathrm{CFU}/\mathrm{mL}=\frac{\left(\mathrm{number}\;\mathrm{of}\;\mathrm{colonies}\right)}{\mathrm{volume}\;\left(0.1\;\mathrm{mL}\right)\times \mathrm{dilution}\;\mathrm{factor}}$$

Furthermore, a live/dead assay (BacLight, bacterial viability kit, Thermo Fisher Scientific, Waltham, MA USA) was also executed. To this end, 20 µL of the bacterial suspensions, prepared as above, were incubated on Si_Ag and Si for 30 min in a humidified atmosphere. After the incubation period, Si_Ag and Si were marked with a 1:1 mix of SYTO^®^ 9 nucleic acid (green fluorescent) and propidium iodide (red fluorescent), and finally, they were observed by confocal microscopy.

## 3. Results and Discussions

### 3.1. Ag Dendrites Synthesis and Characterization

A 3D fractal Ag dendrite antibacterial platform was achieved by metal-assisted chemical etching fabrication, as schematized in Figure 1a,b. In particular, after a typical microelectronics cleaning of a silicon wafer involving 5 min in Acetone, 5 min in Isopropanol, and a rinse in water, the silicon wafer was cleaned from organic contaminants with a 2 min UV ozone treatment. After the cleaning, the native oxide was removed using a 5% HF aqueous solution. After this step, the sample was dipped in a solution containing 0.02 M of silver nitrate (AgNO_3_) as a metal precursor and 5 M of HF diluted in deionized water for 30 min at room temperature. Silver salts dissociate into single anions and precipitate as clusters onto the Si substrate, as shown in Figure 1a. Additional Ag^+^ ions that precipitate onto the substrate cluster onto the initial Ag particle seeds, forming the Ag dendrites, as shown in Figure 1b. During the same process, the Ag particles act as a catalyst for the etching of the silicon under them, causing the fabrication of Si NW [50]. All the processes are carried out at room temperature.

The morphology of the final samples is shown in the SEM images reported in Figure 1c,d. In particular, in Figure 1c, a cross-section SEM image shows the Ag dendrite fractal network with an average thickness of 15 ± 5 µm over an array of Si NWs of about 2 µm in length. In Figure 1d, the plan-view SEM image of the obtained Ag dendrites sample is reported, showing the typical trenches morphology. The SEM analysis revealed a dense forest of Ag dendrites that results in a promising disordered porous network of spiked metal nanostructures of great interest for antibacterial applications.

An EDX analysis has been performed for the as-grown Ag dendrite samples shown in the cross-section SEM in Figure 2a. In particular, in Figure 2b–d, the elemental maps for Si (in green), Ag (in blue), and O (in red), respectively, are reported. In Figure 2e, the profile measurements following the red dashed line visible in Figure 2a are reported, clearly showing the good quality of the Ag dendrites, with a negligible oxygen amount and without any contamination.

For bio-applications, a critical aspect is the removal of organic surface contaminants that can be present due to the storage or handling of the samples. In our previous work on their use as an SERS substrate, we have demonstrated that using our storage protocol and a simple UV-Ozone treatment, we cannot observe any presence of organic contamination [48].

### 3.2. Cytocompatibility Evaluation of hFOB on Si_Ag and Si

To evaluate the cytocompatibility, in terms of cell viability and proliferation, we performed an MTT assay and DAPI staining on hFOB cells grown on Si_Ag and Si for 24, 48, and 72 h. Cell viability results are showed in Figure 3. The MTT assay showed that after 24 h, there is a slight increase in viable cells of about 5% on Si_Ag compared to Si (105 ± 3% vs. 100 ± 0.67%); after 48 h, no significant difference is displayed (97.4 ± 2.56% vs. 100 ± 3.66%); however, after 72 h, there is a reduction of about 17% (82.8 ± 3.85% vs. 100 ± 2.92%).

Similar results were obtained by DAPI staining. Representative DAPI staining images are presented in Figure 4, showing that in Si_Ag, there is a similar number of hFOB cells compared to Si at all analyzed timepoints. Specifically, the cell count showed that in Si_Ag at 24 h, there are about 4.6% more DAPI positive cells, whereas after 48 h and 72 h, there are 5.6% and 6.1% less compared to those in the Si only bulk substrate. In addition, it is possible to note a reduction in cell number for both substrates (Si and Si_Ag) from 24 h to 72 h. Specifically, after 48 h and 72 h, the Si substrate shows a decrease of about 15.9% and 22.2% compared to that at 24 h, respectively, while there is a reduction of about 7.4% between 48 h and 72 h. Similarly, after 48 h and 72 h, Si_Ag shows a decrease of about 24.1% and 30.1% compared to that at 24 h, respectively, while about a 7.9% decrease between 48 h and 72 h.

Data obtained from both MTT and DAPI analyses highlight that from 24 to 72 h, there is a slight, not significant, decrease in cell number between the Si_Ag and Si substrates, while a more marked difference is visible compared to the different timepoints of the Si_Ag substrate. These data compared to other findings reported in the literature on silver-based coatings for biomedical devices suggest that fractal Ag dendrite coating exhibits the best cytocompatibility [22,51]. It has been previously demonstrated that silver nanoparticles are particularly appealing for numerous biomedical purposes, due to the strong antimicrobial activity of released Ag^+^ ions over a prolonged time [52,53]. Unfortunately, the release of Ag^+^ ions induces a series of unpredictable consequences in biological systems, including cell toxicity, genotoxicity, immunologic reactions, and even cell death [54,55,56], and so their use in the biomedical field invokes great worries about human healthy [57]. In addition, despite the broad variety of Ag-NPs applications, to date, their toxic mechanism is still uncertain. Some in vitro and in vivo studies suggested that the cytotoxicity of Ag-NPs depend on their concentration, size, shape, exposure time, and environmental factors [58]. Other evidence showed that Ag-NPs can interact with cell membrane proteins and generate reactive oxygen species (ROS), causing damage of proteins and nucleic acids and ultimately, causing cell proliferation inhibition and apoptosis [59].

### 3.3. Antibacterial Assay of Ag Dendrites

To estimate the antibacterial activity of Si_Ag on *S. aureus* and *P. aeruginosa* strains, a viability assay after 24 h of bacteria incubation on both Si and Si_Ag substrates was performed for 72 h. The results, reported in Figure 5, show no toxic effect of the Si substrate on the growth of either bacteria suspensions. Conversely, the Si_Ag substrate shows a significant inhibiting activity on the growth of both bacterial strains, as highlighted by an important CFU decrease after 24 h of incubation, even if the antibacterial activity is more evident for *P. aeruginosa* than for *S. aureus*. More specifically, the CFU number for *P. aeruginosa* was about 2.1 ± 0.7 × 10^5^ CFU/mL on Si_Ag and 6.5 ± 1.1 × 10^9^ CFU/mL on Si substrate, indicating a decrease of 99.997% for Si_Ag compared to Si. On the other hand, *S. aureus* exhibited a reduction of about 97.5%, 2 ± 0.6 × 10^8^ CFU/mL, on Si_Ag compared to Si (8 ± 1.2 × 10^9^ CFU/mL).

These finding should result from the different structural organizations of the cell wall in Gram-negative (*P. aeruginosa*) compared to Gram-positive (*S. aureus*) [60,61,62,63]. It has been established that metal NPs can interact with the repetitive units of the amino acids and carbohydrates of the peptidoglycan coating; therefore, since Gram-positive walls contain a larger quantity of peptidoglycan than Gram-negative walls, these would be more resistant to damage [64]. The Gram-positive walls should also favor resistance against metal ions, which would be trapped by the greater amount of negatively charged peptidoglycan [62]. On the other hand, the higher sensitivity of Gram-negative to AgNPs could also be due to external lipopolysaccharide (LPS) and its peroxidation by reactive oxygen species (ROS).

Recent studies suggest that the LPS mosaic conformation in Gram-negative bacteria make some areas more negatively charged, and so positively charged AgNPs tend to cluster in these regions, causing a limited lethal effect [65]

Our findings were also confirmed by a live/dead BacLight bacterial viability kit evaluating the bacteria deposited on the Si_Ag and Si scaffolds (Figure 6). The Si scaffolds, without silver fractal-like structures, exhibited a great density of live cells (green stains) for both bacterial strains (Figure 6), according to the green fluorescent living cells (Syto^®^ 9); contrarily, most cells are dead in the Si_Ag substrate, as showed by red fluorescence (propidium iodide).

In contrast, scaffolds loaded with Ag dendrites (Si_Ag) showed a strong surface biocidal property, as indicated by the presence of dead bacteria (red stains) and rare areas of live bacteria (green stains). From fluorescence images, it was also possible to locate Ag dendrites (dull green) on the surface of the silicon slides. In this regard, a different susceptibility of the two tested bacterial strains could be again noted. In fact, regarding *S. aureus*, we observe a higher cell density on the silver fractal-like structures. Rare live cells can be seen on these structures, although most of them are dead. Contrastingly, for *P. aeruginosa*, a lower density of cells, all dead, near the Ag dendrites was observed. These results would confirm what was previously indicated by the CFU assays.

## 4. Conclusions

In this paper, we have evaluated the biological response of human and bacterial cells on silicon substrates coated with fractal silver dendrites (Si_Ag) for use in bone tissue engineering applications. Ag dendrites have been produced by a simple chemical deposition method realized by an industrially compatible silicon-based process for the fabrication of an Ag dendrite fractal network for antibacterial applications. Cytocompatibility and bacteria viability of hFOB cells and *P. aeruginosa* and *S. aureus* bacteria, respectively, have been performed to evaluate the cytotoxic effect and antibacterial activity of this coating. Data collected from MTT and DAPI analyses indicated that hFOB cells cultured on the Si_Ag surface display a decrease in cell number of approximately 20–30% over time, suggesting that the fractal Ag dendrite coating exhibits the best cytocompatibility compared to other silver-based coatings reported in the literature. In addition, results from bacterial viability and live dead staining displayed a significant reduction in bacteria after 24 h of incubation on Si_Ag for both strains, more evident for *P. aeruginosa* than for *S. aureus*, indicating the remarkable antibacterial activity of this coating. Therefore, these findings taken together suggest that fractal silver dendrite nanostructures could represent an eligible nanomaterial for the coating of implantable medical devices.

## Figures and Tables

**Figure 1 nanomaterials-13-01107-f001:**
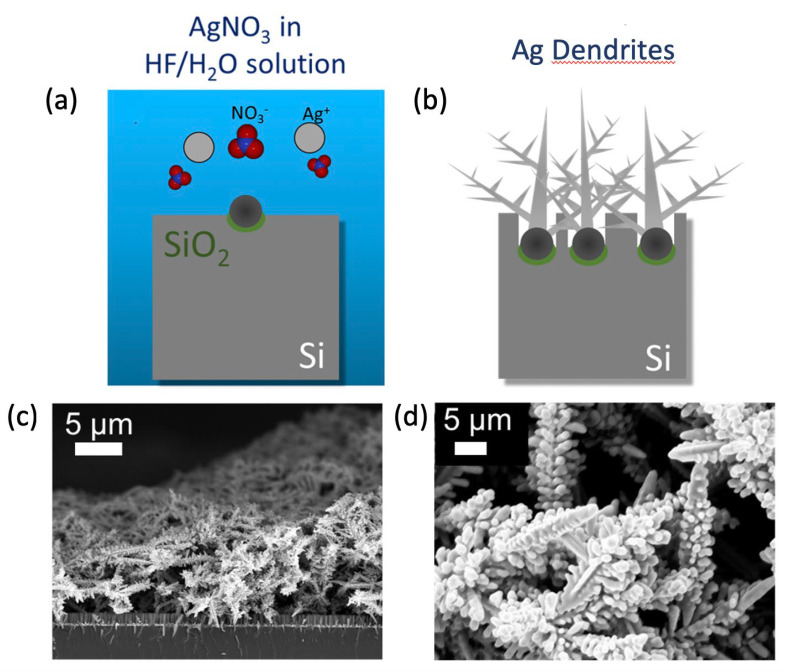
Ag dendrites fabrication. Schematic representation of the Ag dendrite fabrication protocols: (**a**) AgNO_3_ salts dissolution in HF/H_2_O leading to the precipitation of Ag nanoparticles onto the Si bulk substrate; (**b**) Ag dendrite formation due to Ag^+^ ions clustering onto the initial Ag particle seeds. Reprinted from ref. [48]. SEM images of the Ag dendrite samples in cross-section and in plan-view are reported in (**c**) and (**d**), respectively.

**Figure 2 nanomaterials-13-01107-f002:**
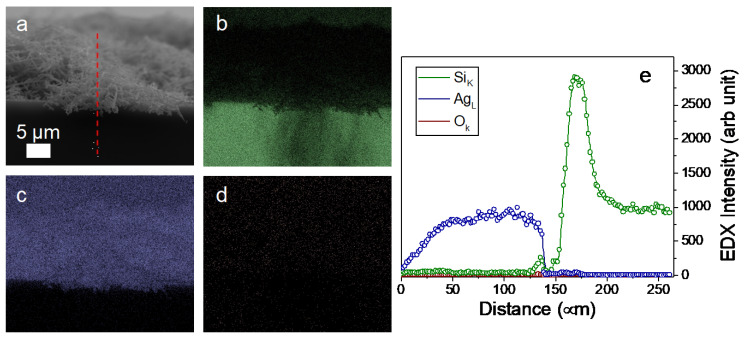
Compositional map of the Ag dendrite samples obtained by energy dispersive X-ray spectroscopy (EDX). (**a**) SEM image in cross configuration of the sample. (**b**) Silicon map (in green), (**c**) Ag map (in blue), and (**d**) O map (in red). (**e**) EDX profilometry elemental analysis for silicon, silver, and oxygen performed along the red dashed line shown in (**a**).

**Figure 3 nanomaterials-13-01107-f003:**
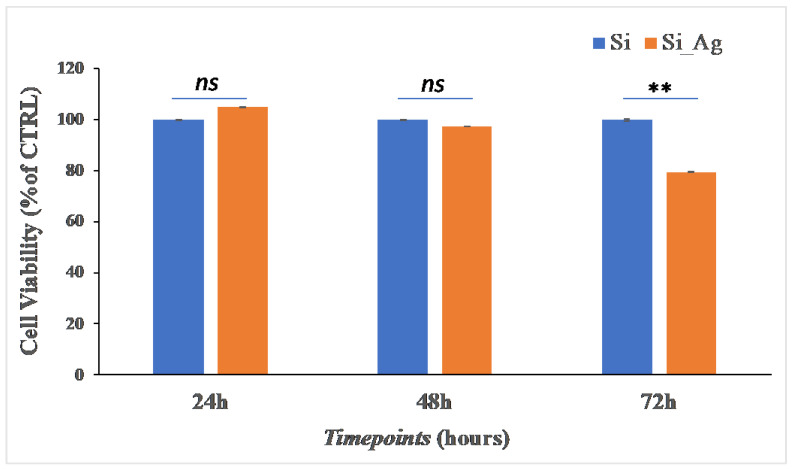
Cell viability. MTT analysis on hFOB grown on the surface of Si_Ag for 24, 48, and 72 h compared to the control (Si). Data are reported as mean ± standard deviation obtained from 3 different samples. ** *p* < 0.01 shows significant differences between the two samples to the same timepoint, as reported by a *t*-test; ns = not significant.

**Figure 4 nanomaterials-13-01107-f004:**
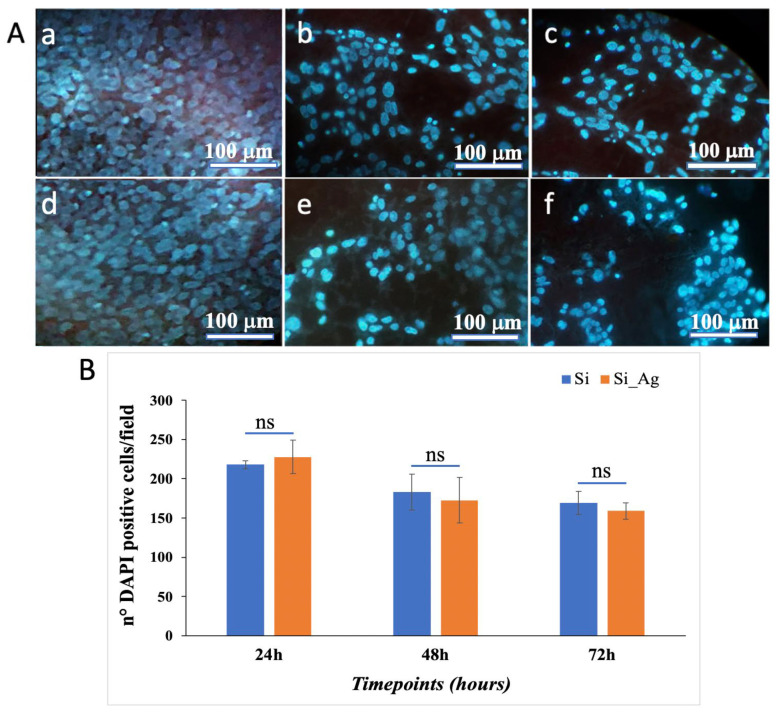
Cell proliferation. (**A**) Representative DAPI staining images of hFOB grown on the surface of the Si bulk substrate (**a**–**c**) and Ag dendrites (**d**–**f**) for 24, 48, and 72 h. (**B**) Graphical representation of DAPI positive cells/field. Data are reported as mean ± standard deviation obtained in 4 different fields; ns = not significant.

**Figure 5 nanomaterials-13-01107-f005:**
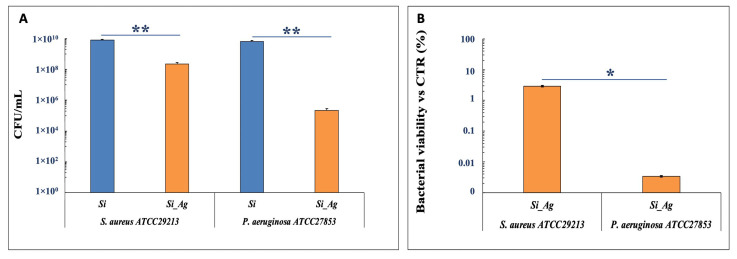
Antibacterial activity of Si_Ag. (**A**) CFU number of *S. aureus* ATCC29213 and *P. aeruginosa* ATCC27853 in the presence of Si and Si_Ag substrates. (**B**) % Bacterial viability vs CTR of *S. aureus* ATCC29213 and *P. aeruginosa* ATCC27853 in the presence of Si_Ag substrate. Data are reported as mean ± standard deviation obtained on 4 different samples. ** *p* < 0.01 and * *p* < 0.05 show significant differences between experimental groups, as reported by a *t*-test.

**Figure 6 nanomaterials-13-01107-f006:**
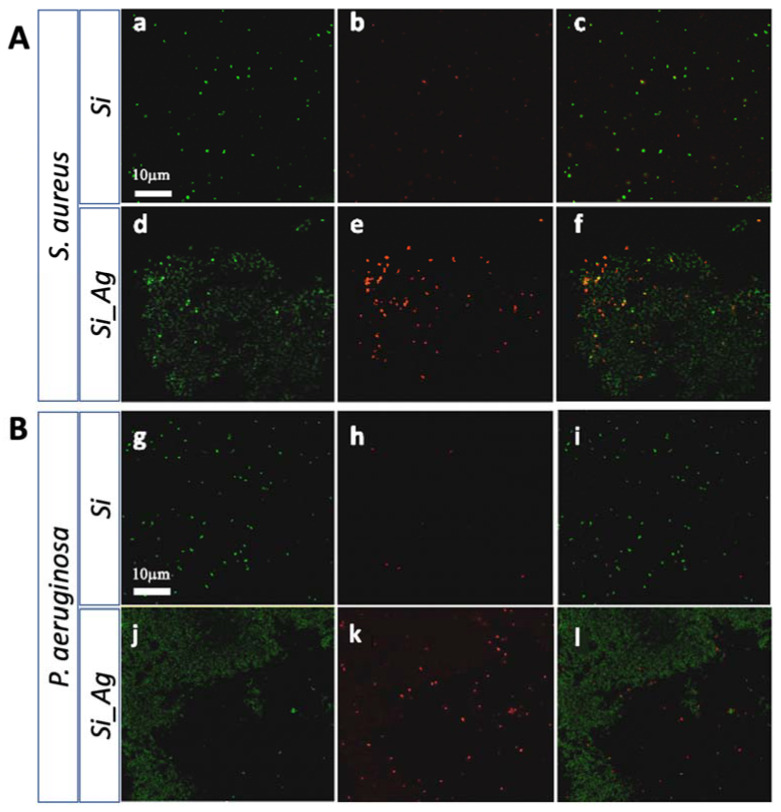
Live/dead staining. (**A**) Representative images of *S. aureus* ATCC29213 on the Si (**a**–**c**) and Si_Ag (**d**–**f**) substrates. (**B**) Representative images of *P. aeruginosa* ATCC27853 on the Si (**g**–**i**) and Si_Ag (**j**–**l**) substrates. Live cells are shown in green and dead cells in red.

## Data Availability

The data presented in this study are available on request from the corresponding author.

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
