# Peer review of "Biological Response Evaluation of Human Fetal Osteoblast Cells and Bacterial Cells on Fractal Silver Dendrites for Bone Tissue Engineering"

_nanomaterials, 2023, doi:10.3390/nano13061107_

Round 1

Reviewer 1 Report

The authors have prepared fractal silver dendrites substrates by silicon-based technology and then evaluated their biological response of hFOB, P. aeruginosa and S. aureus. In vitro study, The Si_Ag surface has revealed good cytocompatibility for at least 72 h. The Si_Ag surface has also decreased pathogens viability for both gram-positive and gram-negative. Overall, this work can inspire more material design ideas of silver-based nanomaterials for bone tissue engineering. Therefore, I would like to recommend this work to publish in Nanomaterials. Below are some comments for the authors.

1. This paper would be more impressive if the authors could provide the Energy Dispersive X-Ray analysis (EDX) of Ag dendrites in Figure 1c or 1d.

2. For Figure 3A, the scale bar should be added in each image.

3. For Figure 5, the right column should be described in caption. Moreover, for caption of Figure 5, the authors described “In green are shown live cells and in red”, what is in red? Please correct.

4. For the introduction “Several advanced nanomaterials displaying antibacterial activity have broadly reported in literature, including metal ions and metal oxide nanoparticles”, more references could be cited to broaden the introduction.

https://doi.org/10.2147/IJN.S328767

Author Response

Reviewer 1

The authors have prepared fractal silver dendrites substrates by silicon-based technology and then evaluated their biological response of hFOB, P. aeruginosa and S. aureus. In vitro study, The Si_Ag surface has revealed good cytocompatibility for at least 72 h. The Si_Ag surface has also decreased pathogens viability for both gram-positive and gram-negative. Overall, this work can inspire more material design ideas of silver-based nanomaterials for bone tissue engineering. Therefore, I would like to recommend this work to publish in Nanomaterials. Below are some comments for the authors.

  1. This paper would be more impressive if the authors could provide the Energy Dispersive X-Ray analysis (EDX) of Ag dendrites in Figure 1c or 1d.

Response: We acknowledge the referee for his/her careful reading and for his/her comments. We have performed an EDX analysis of the Ag dendrites as requested. In particular, we have both performed an EDX mapping of the silicon, silver and oxygen content and a profilometry analysis in cross-section. Moreover, by the EDX we can further exclude the presence of contaminants on the samples surface. We added the Figure 2 and the following comment into the manuscript in Results and discussions, paragraph 3.1. Ag dendrites synthesis and characterization.

“An EDX analysis has been performed for the as-grown Ag dendrite samples shown in the cross-section SEM in Figure 2 a. In particular, in Figure 2 b-d the elemental maps for Si (in green), Ag (in blue), and O (in red), is respectively reported.  In Figure 2 e the profile measurements following the red dashed line visible in Figure 2a is reported clearly showing the good quality of the Ag dendrites with a negligible oxygen amount and without any contamination.”

  1. For Figure 3A, the scale bar should be added in each image.

Response: We thanks the reviewer for his/her comments. We added the scale bar in each image of the Figure 3A, now Figure 4A.

  1. For Figure 5, the right column should be described in caption. Moreover, for caption of Figure 5, the authors described “In green are shown live cells and in red”, what is in red? Please correct.

Response: We thanks the reviewer for his/her comments. We added the description in caption of Figure 5, now Figure 6 and we modified the sentence “In green are shown live cells and in red” with “Live cells are shown in green and dead cells in red. Inserts are magnifications of the drawn areas”

  1. For the introduction “Several advanced nanomaterials displaying antibacterial activity have broadly reported in literature, including metal ions and metal oxide nanoparticles”, more references could be cited to broaden the introduction.

https://doi.org/10.2147/IJN.S328767

     Response: We thanks the reviewer for his/her comments. We added the sentence “The antibacterial properties of these metal ions and metal oxide NPs are mainly due to their ability to release metal ions in aqueous media, that result extremely toxics against bacteria [21]” in Introduction section, reporting the reference suggested.

Reviewer 2 Report

The article of " Biological response evaluation of human fetal osteoblast cells and bacterial cells on fractal silver dendrites for bone tissue engineering” is an interesting topic. Several areas of the article need to be improved.

1.      Line 137: what is the MACE protocol?

Line 167: What is the PFA? ……

It is necessary to write the abbreviation first and put the full name in brackets after it at first mention.

2.      How to use the silicon substrate coated with fractal silver dendrites in bone tissue engineering?

3.      The size of cellular nuclei became small over the time course after 24 hours incubation on Si and Si_Ag in Figure 3a. The author needs to explain this phenomenon.

4. The quality of Figure 5 is not well.

Author Response

Reviewer 2

The article of " Biological response evaluation of human fetal osteoblast cells and bacterial cells on fractal silver dendrites for bone tissue engineering” is an interesting topic. Several areas of the article need to be improved.

  1. Line 137: what is the MACE protocol?

Response: We thank the referee for his/her careful reading. The Metal Assisted Chemical Etching (MACE) protocol [Nanomaterials 2021, 11 (2), 383] is a widely diffused approach for the cost-effective realization of Si NWs already adopted in several research labs. MACE approach is based on a wet etching process driven by a metal catalyst deposited onto the surface of the silicon. The process is typically at room temperature and no metal contamination is attested in the final nanostructures. In particular, the silver salt approach used to realize these Ag dendrites has been the first MACE method demonstrated in 2002 by Peng et al. [Adv. Mater. 2002, 14, 1164–1167].

We modified the paragraph 2.2 “Ag dendrites structural characterizations” in material and methods section, to add a brief explanation “.

Line 167: What is the PFA? ……

It is necessary to write the abbreviation first and put the full name in brackets after it at first mention.

Response: We thank the referee for his/her careful reading. PFA is the acronymous of Paraformaldehyde. We’ve put the full name in brackets after it at first mention, in Material and methods, paragraph “2.4. Cells viability and proliferation on fractal silver dendrites”.

  1. How to use the silicon substrate coated with fractal silver dendrites in bone tissue engineering? 

Response: We thank the referee for the interesting question. The fractal silver dendrites can be removed from the Si substrate by a ultrasonication bath and thus, adopted independently to the starting substrate. We used silicon substrate only for in vitro evaluation, but our idea is to coat medical devices, for example titanium/ ceramic prostheses to use in bone tissue engineering.

  1. The size of cellular nuclei became small over the time course after 24 hours incubation on Si and Si_Ag in Figure 3a. The author needs to explain this phenomenon.

Response: We thank the referee for the interesting question. Sorry for the typo but we have inserted in Figure 3a e 3d a different magnification compared to Si and Si_Ag. We have modified the image in the revised version of the manuscript (new figure 4a e 4d).

  1. The quality of Figure 5 is not well.

Response: We thank the referee for the interesting question. Sorry for the poor quality of Figure 5. We tried to improve the image quality modifying the pixels from 300dpi to 600dpi (new figure 6).

Reviewer 3 Report

The concept of using silver dendrites for surface coating of devices is novel. However, the reviewer has some questions

1) This dendrites seem to be of macroscopic size with sharp edges. I wonder if this sharp needle like edges are used as implant coating especially in regions where there is mobility like bone; wont it pierce through the membrane or surface and create abrasions? The cell death could be because of he pi Also I am unsure what will be the effect of irregular dendrite surface in the bio-environment. A zoomed in visual of each dendrite and reproducibility of each dendrite structure will be important.

2) Kindly include references of dendrites used as coating implants

3) It is not clear how does generating a dendrite structure help in overcoming the Ag toxicity. Only the structure of silver is changed but molecularly silver remains the same. Kindly elaborate. 

Author Response

Reviewer 3

The concept of using silver dendrites for surface coating of devices is novel. However, the reviewer has some questions.

  • This dendrites seem to be of macroscopic size with sharp edges. I wonder if this sharp needle like edges are used as implant coating especially in regions where there is mobility like bone; wont it pierce through the membrane or surface and create abrasions? Also I am unsure what will be the effect of irregular dendrite surface in the bio-environment.

Response: We thank the referee for the interesting question. We think that this sharp needle like edges don’t should pierce through the membrane or surface and create abrasions, as demonstrated by several evidences reported in literature for biomedical application (i.e. “Functional Dendritic Coatings for Biomedical Implants Emerging Trends in Nanomedicine”), but to verify this possibility other in vivo studies will be performed following. Moreover, in the manuscript we already reported several biological tests involving cell viability and proliferation and we didn’t note any negative effects for the bio-environment, as showed in Results section, paragraph 3.2. Cytocompatibility evaluation of hFOB on Si_Ag and Si.

The cell death could be because of he pi?

We think that the reviewer refers to pH or pHI, this is a good question and we in the next in vitro studies will address this point.

A zoomed in visual of each dendrite and reproducibility of each dendrite structure will be important.

These Ag dendrites are characterized by a fractal structure that some of the authors have already studied and demonstrated in the following reference [45]. The synthesis process is a well diffused and reliable method. Moreover, the fractal morphology ensures a reproducibility of the same random morphology on more than one observation scale.

  • Kindly include references of dendrites used as coating implants.

Response: We thank the referee for his/her suggestion. We added the references of dendrites used as coating implants [30,31], in revised version of manuscript.

  • It is not clear how does generating a dendrite structure help in overcoming the Ag toxicity. Only the structure of silver is changed but molecularly silver remains the same. Kindly elaborate. 

Response: We thank the referee for his/her comment. We think that this can be attributed probably because by the MACE protocol the silver ions precipitate in the substrate forming dendrites and the Ag ions do not remain in solution, as instead happens for the Ag NPs.

Round 2

Reviewer 2 Report

The authors have answered all of my questions and the paper has been greatly improved. Therefore, it can be accepted for publication.

Reviewer 3 Report

The authors have responded to the queries raised satisfactorily. The paper can be accepted.